# A 3D Keypoints Voting Network for 6DoF Pose Estimation in Indoor Scene

**Huikai Liu** [1,2,3]**, Gaorui Liu** [1,3]**, Yue Zhang** [1,2,3]**, Linjian Lei** [1,3,4]**, Hui Xie** [1,2,3]**, Yan Li** [1,2,3] **and Shengli Sun** [1,3,*]

1  Shanghai Institute of Technical Physics, Chinese Academy of Sciences, Shanghai 200083, China;
   liuhuikai@mail.sitp.ac.cn (H.L.); kingrain@mail.ustc.edu.cn (G.L.); zhangyue8@mail.sitp.ac.cn (Y.Z.);
   leilj@shanghaitech.edu.cn (L.L.); xiehui@mail.sitp.ac.cn (H.X.); liyan@mail.sitp.ac.cn (Y.L.)
2  School of Electronic, Electrical and Communication Engineering, University of Chinese Academy of Sciences,
   Beijing 100049, China
3  Key Laboratory of Intelligent Infrared Perception, Chinese Academy of Sciences, Shanghai 200083, China
4  School of Information Science and Technology, ShanghaiTech University, Shanghai 201210, China
*  Correspondence: palm_sun@mail.sitp.ac.cn

**Abstract:** This paper addresses the problem of instance-level 6DoF pose estimation from a single RGBD image in an indoor scene. Many recent works have shown that a two-stage network, which first detects the keypoints and then regresses the keypoints for 6d pose estimation, achieves remarkable performance. However, the previous methods concern little about channel-wise attention and the keypoints are not selected by comprehensive use of RGBD information, which limits the performance of the network. To enhance RGB feature representation ability, a modular Split-Attention block that enables attention across feature-map groups is proposed. In addition, by combining the Oriented FAST and Rotated BRIEF (ORB) keypoints and the Farthest Point Sample (FPS) algorithm, a simple but effective keypoint selection method named ORB-FPS is presented to avoid the keypoints appear on the non-salient regions. The proposed algorithm is tested on the Linemod and the YCB-Video dataset, the experimental results demonstrate that our method outperforms the current approaches, achieves ADD(S) accuracy of 94.5% on the Linemod dataset and 91.4% on the YCB-Video dataset.

**Keywords:** 6DoF pose estimation; split-channel attention; ORB-FPS keypoint

## 1. Introduction

6d pose estimation is a functional task of many computer vision applications, such as augmented reality [1], autonomous navigation [2,3], robot grasping [4,5] and intelligent manufacturing. The purpose of 6d pose estimation is to obtain the rotation and translation from the object's coordinate system to the camera's coordinate system. In practical applications, the estimation process requires robustness to the noise, occlusion, different lighting conditions and achieves real-time. RGB image has rich texture information, so the pose of the object can be estimated by detecting the object in the images. Traditional RGB-based methods [6] extract the global or local feature to match the source model. With the rapid development of Convolutional Neural Network (CNN), image feature learning ability has been significantly improved. CNN is also used in pose estimation, PVNet [7] regress the 2d keypoint through the end-to-end network, and then use the PnP algorithm, estimate the 6d pose by calculating the 2d-3d correspondence relationship of the object. RGB-based methods always achieve good computing efficiency, but they are sensitive to the background, illumination, texture. In addition, these RGB-based approaches need to compute the projection of the 3d model, making partly lose of the geometry constraints. Depth image or point cloud contains sufficient geometry information, Hinterstoisser et al. [8] uses hand-crafted feature or pointnet to extract the point cloud feature and estimate the pose by feature matching or bounding box. Compared to RGB images, the point cloud has more geometry information. However, the point cloud does not have texture information and it

is sparse. In addition, due to the mirror reflection, RGB-D sensors cannot obtain the depth information of the smooth or transparent surface.

Based on the limitations mentioned above of RGB-based methods and pointcloud-based methods, how to fully use the RGB image's texture information and pointcloud's geometry information becomes an important issue. PoseCNN [9] first uses RGB image to estimate the coarse pose of the object, then utilizes Iterative closest point(ICP) [10] to refine the result. However, these two separate steps cannot be optimized jointly and are time-consuming for real-time applications. Chi et al. [11] extracts the depth image's information through CNN and uses it as a supplementary channel of the RGB image, but it needs a complex preprocess and does not make the best use of the RGBD information. DenseFusion [12] is an innovative work, it extracts the RGB and point cloud information from every single pixel by CNN and PointNet [13], respectively, then embeds and fuse the RGB and point cloud information of every pixel through the single-pixel fusion method. The later method FFB6D [14] adds the communication between the two channels. Full pixel estimation greatly increases the computed complexity, PVN3D [15] uses full pixel hough voting to obtain the 3d keypoints and then estimate the 6d pose through the least-squares method. Compared to 2d-keypoint-based methods, PVN3D significantly increases the robustness. However, PVN3D gets the keypoints by farthest point sample (FPS), which only concerns the Euclidean distance factor, without the texture information which may cause the selected keypoints appear on a smooth surface.

In order to make comprehensive use of image and point cloud information, we propose the 3D keypoint voting network (3DKV) for 6DoF pose estimation. As shown in Figure 1, 3DKV is a two-stage network, which first detects the keypoints, then utilizes the least-squares method to compute the 6d pose. To enhance RGB feature representation ability, we present the feature map split-attention block. More specifically, every block divides the feature map into several groups along the channel dimension and finer-grained into subgroups, and the group is determined via a weighted combination of its subgroups, where the weights are calculated through the contextual information. In addition, an ORB-FPS keypoint election approach that concerns both texture and geometry is proposed. Firstly, we detect the ORB keypoint in the RGB images, then calculate the correspondence 3d points in the point cloud through the camera parameter, then find the final 3d ORB-FPS keypoints through the Farthest Points Sample (FPS) from the selected points. This method improves the ability of keypoints to characterize objects, and avoids the selected keypoints appear on non-significant areas like smooth surfaces, making it easier to locate keypoints and improving the ability to estimate the pose. In general, the contributions of this paper can be concluded as follows:

(1) A split-attention block is presented in the image feature extraction part. This method combines channel-attention and group convolution. The channel-attention block enhances the feature fusion between the image's channel dimension. Moreover, the group convolution reduces the network's parameters and improves its computational efficiency.

(2) A simple and effective keypoints selection approach named ORB-FPS is proposed. It utilizes a two-stage approach to select the keypoints, and avoids the selected keypoints appear on non-significant areas like smooth surfaces, making them easier to locate and improving the network's ability to estimate the pose.

(3) A thorough evaluation of the proposed algorithm is presented, including comparisons with the current algorithms on the Linemod and YCB datasets. The experiment results show that the proposed method performs better than other algorithms.

The remainder of this paper is organized as follows. Section 2 reviews the related work of other researchers. Section 3 demonstrates the details of our proposed method. Section 4 provides the experiment results and analyses. Section 5 concludes with the summary and the perspectives.

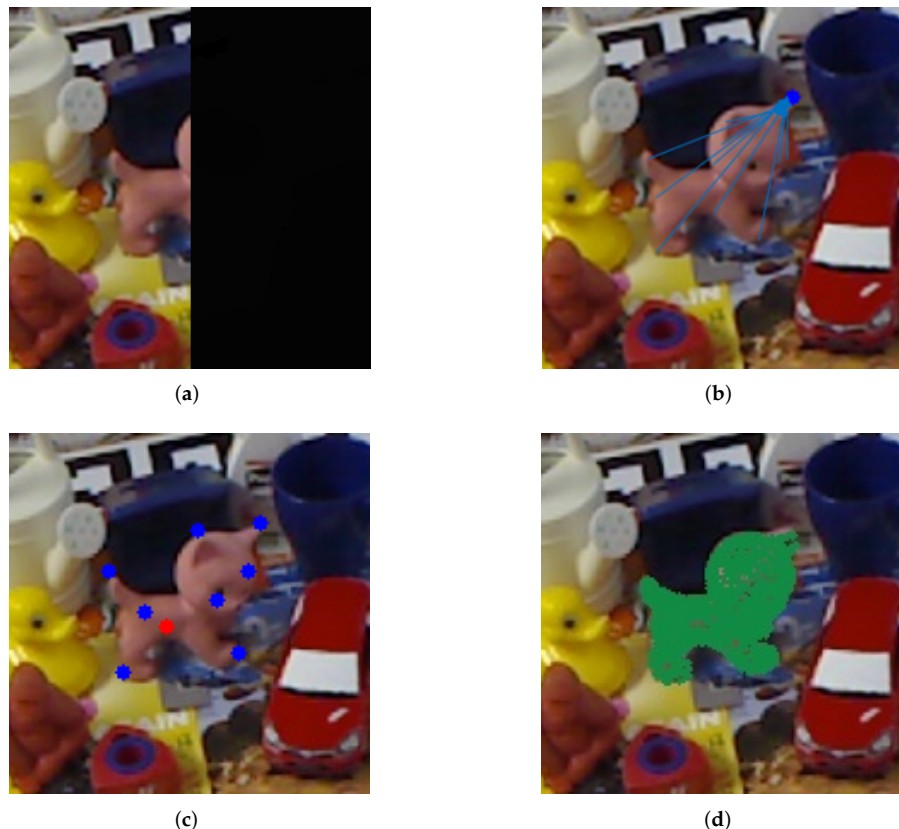

**Figure 1.** Pipeline of our proposed algorithm. (**a**) The RGB-D input data of our network; (**b**) The blue lines represent the direction vector to the keypoint; (**c**) The predicted 9 points, where 8 blue ORB-FPS keypoints and one red center point; (**d**) The 6DoF pose estimated by our algorithm.

## 2. Related Work

### 2.1. Pose from RGB Images

The RGB image-based methods for 6DoF pose estimation can be roughly divided into three categories: template, correspondence relationship and voting. The template-based approaches first learn the feature of objects through convolutional neural networks, and then detect the objects in the image and calculate the pose. SSD6D [16] extends the SSD detection network to enable instance segmentation and 6d pose estimation. PoseCNN [9] proposes an end-to-end method based on RGB images, it includes three modules: semantic segmentation, 3D translation estimation and 3D rotation estimation, and uses the VGG16 to extract features. Zeng et al. [17] proposes a unsupervised pose estimation network based on the multi-perspective images. Another keypoint-based strategy first detects the keypoints of the corresponding object from the image and then calculates the pose by regressing the keypoints. For example, YoLo6D [18] detects the center point and 8 bounding box points of the projection of the three-dimensional object on the image, then utilizes the PnP algorithm to obtain the final pose. DpoD [19] uses the dense UV map to directly obtain the connection between the 2d pixel and the vertex of the 3D model. Some other methods vote on the pixels or patches of the image to obtain key points. PVNet [7] first votes the keypoints through RANSAC, then utilizes the 2D-3D correspondence to calculate the 6D pose. HybridPose [20] adds edge vector and symmetrical correspondence into the PVNet framework to enhance the robustness of symmetrical objects. Based on the shortcoming of the PnP algorithm and regress directly, GDR-Net [21] presents a geometry-guided regression Network. Furthermore, Stevsic et al. [22] adds the attention blocks into the feature extraction module to improve its feature representation ability. Some other researchers use panoramic images to accomplish this task, Zhang et al. [23] show how the use of panoramic images improves significantly the geometric analysis of the scene thanks

to the large captured context and Xu et al. [24] it with object poses. Pintore et al. [25] combines modern CNN networks, 3D scene information and parameter optimization to recover the oriented bounding boxes of the captured objects inside the 3D layout of the indoor environment. RGB-based approaches always are efficient, but most of them built on the perspective projection, which may cause the partly lose of the geometric constraint information.

### 2.2. Pose from Point Cloud

With the development of terrestrial laser scanners (TLS) and low-cost 3D scan instruments like Kinect, point clouds can be obtained easily. There are many classical pointcloud-based algorithms for pose estimation, such as calculating FPFH [26], SHOT [27] and other local descriptors. Pointnet [13,28] and its variants make a break work, which directly applies deep learning to point clouds and enable them to complete advanced applications like object classification, semantic segmentation, object recognition. A variety of 6d pose estimation approaches are also proposed based on PointNet, Votenet [29] uses Hough voting to generate points close to the center of the object, then group and aggregate the points to obtain candidate boxes. PointnetLK [30] expands the PointNet and Lucas–Kanade (LK) algorithms into a single trainable recurrent deep neural network and achieves outstanding pose estimation performance. Weng et al. [31] proposes category-level 9Dof pose estimation, which consists of 3D rotation, 3D translation and 3D size. Huang et al. [32] predicts the pose by learning the stable geometry feature. PCRNet [33] uses PointNet as the backbone to extract the global feature for pose estimation, it is more robust to noise. Gao et al. [34] proposes a lightweight data synthesis pipeline to produce the data. However, the pointcloud is textureless and sparse. Meanwhile, due to the specular reflection, the depth camera cannot obtain the depth information of the smooth or transparent surface, which will limit the performance.

### 2.3. Pose from RGBD Data

Based on the above-discussed shortcomings of RGB-based and pointcloud-based approaches, some researchers combine the two types of information together. Traditional methods for pose estimation mainly use hand-crafted features. For example, Linemod [8] locates and estimates object poses by extracting gradient features of images and normal features of depth images. Some other methods use deep learning to extract the RGBD feature, Shao et al. [35] proposes two fusion strategies, the first is concatenates RGB and depth image into a raw input to the CNN network, and another strategy just like [3,36,37], they utilize CNN network to extract the RGB image and depth image features, and then concatenate the features as the input for object segmentation and pose estimation. However, these methods neglects the inner structure of the depth channel and extract depth image features as a supplement channel to the RGB feature channels. Densefusion [12] separately extracts RGB and depth feature information through CNN and PointNet, and designs a dense pixel-level fusion method, which integrates the features of RGB data and point cloud features more properly. In order to enhance the connection between the two channels, refs. [14,38] built the full flow bidirectional fusion and correlation fusion communication module. PVN3D [15] proposes a new method to generate 3D keypoints. It generates 3D key points through full-pixel voting and calculates 6D pose by using the least square method, 6-PACK [39] proposes an anchor-based attention network to generate an order of keypoints. 3D keypoints strategy improves the pose estimation performance significantly, however, it only uses the distance factor and the RGB texture information is not effectively used.

## 3. Methodology

Given an RGBD image, 6D pose estimation can be described as finding the best affine transformation between the object's coordinates and the camera's coordinate, which consists of two parts: rotation matrix and translation vector. The entire process should be fast and accurate, and robust to noise and occlusion.



In order to tackle this problem, this paper proposes a 3D Key Point Voting (3DKV) strategy for 6d pose estimation. As shown in Figure 2, 3DKV is a two-stage structure, first detects the 3D ORB-FPS keypoints, then predicts the 6D pose. In detail, given an RGB-D image as input, we use an ImageNet pre-trained in-channel attention block CNN network to extract the RGB image's texture and other appearance features and use the PointNet++ [28] to extract the point cloud's geometric features. We add the split-attention blocks into every channel of the image feature extraction module to enhance the channel information fusion. Then 3DKV utilizes DenFusion to fuse the RGB and point cloud feature of every pixel and sends them to the keypoints detection module. In order to make better use of the texture feature, we first detect the 2D ORB keypoints of the RGB image, then finds the 3D fps keypoints from the 3D corresponding points of the 2D orb keypoints. The keypoints detection module includes two parallel tasks: instance semantic segmentation and keypoints prediction. In addition, we select 12,888 points from each RGB-D image as the seed points and use shared Multi-Layer-Perceptron(MLP) to share the training parameters between the 3D keypoint prediction and instance semantic segmentation.

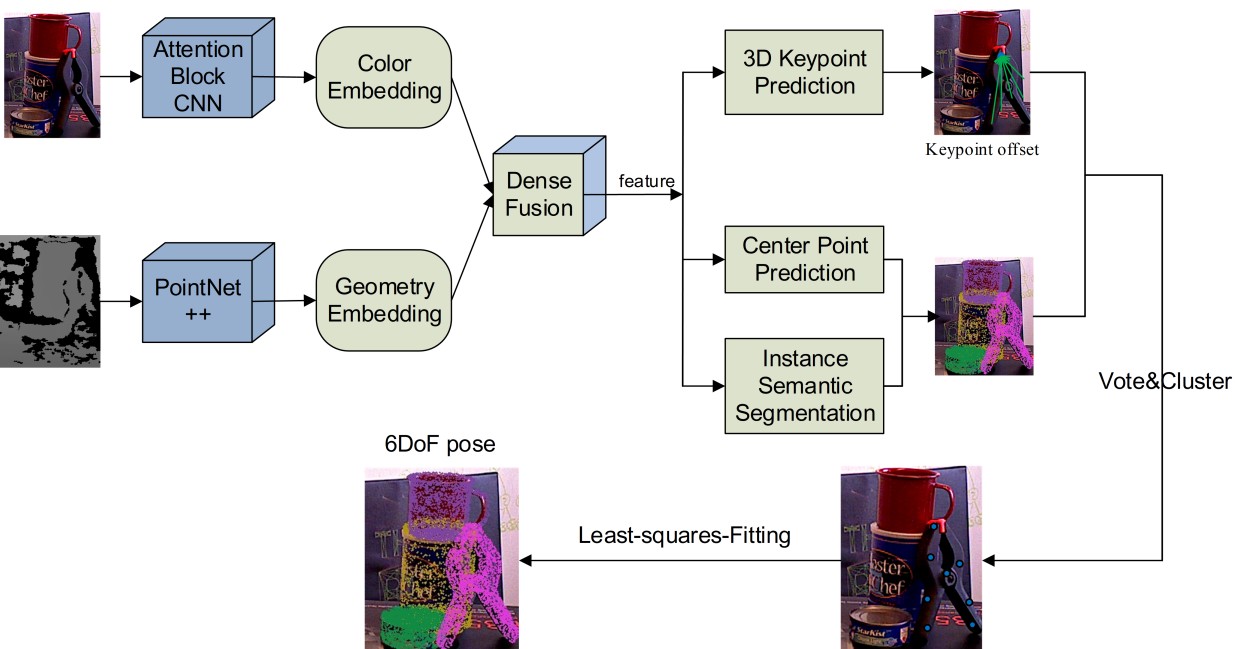

**Figure 2.** Overview of our algorithm. Given RGB image and point cloud, the point-wised features are extracted by the attention CNN network and PointNet++ respectively. The 3D keypoint prediction, center point prediction and instance semantic segmentation modules are trained jointly to obtain the keypoints. Finally, the 6DoF pose is estimated through the least-square algorithm.

### 3.1. Split-Attention for Image Feature Extraction

For the extraction of RGB image features for classification or instance segmentation, the receptive field and the interaction across channels is very important. The convolution operation is the core of the CNN network and its essence is to perform feature fusion on a local area, including spatial (h,w) and cross-channel. Enhancing the network's feature extraction ability can improve its ability of object classification, recognition, segmentation and other applications. Enlarging the receptive field is a common method, that is, fuse more features in spatial or extract multi-scale spatial information [40]. SeNet [41] pays attention to learn the importance of different channel features, which is equivalent to adding attention operations to the channel dimension. SeNet pays more attention to the channel with the most information, while suppresses those that are not important. SkNet [42] designs a dynamic selection mechanism so that every neuron can select different receptive fields according to the target object's size. Based on the above two improvements, Resnest [43]

proposes split-attention, which combines the dynamic receptive field and the attention mechanism to improve the network's ability to express features. Based on the Resnest, we propose the Multi-branch-attention RGB feature extraction module. As shown in Figure 3, given an RGB image with $(h, w, c)$, where $h$, $w$ represent the height and width of the input image and $c$ is the channels. First divide the input channels into $M$ cardinals, denoted as cardinal $1 - M$, and then divide each cardinal into $N$ groups. So the total number of groups is $MN$, suppose the input of each group is $I_i$, then the input of each cardinal can be obtained as:

$$I^m = \sum_{j=N(m-1)+1}^{Nm} (I_j)(m = 1, 2, \ldots M) \tag{1}$$

Since the multi groups of each single cardinal, an average pooling layer is added after each convolution operation, so the weight $W$ of each stream can be calculated as:

$$W^m = \frac{1}{H \times W} \sum_{i=1}^{H} \sum_{j=1}^{W} I^m(i, j) \tag{2}$$

According to the groups weight, the output of every cardinal will be:

$$O^m = \sum_{i=1}^{N} a_i^n I^m(n - 1) + i \tag{3}$$

where $a_i^m$ is the weight after softmax:

$$a_i^m = \{ \begin{array}{ll} \exp W_i^c(w^m) / \sum^{j=0N} \exp W_i^c(w^m) & N > 1 \\ 1/1 + \exp -W_i^c(w^m) & N = 1 \end{array}$$

where $W$ is determined by number of the group in the cardinal. When $N = 1$ means that each cardinal is regarded as a whole. After obtaining the output of each cardinal, the final output of these cardinals can be obtained through splitting.

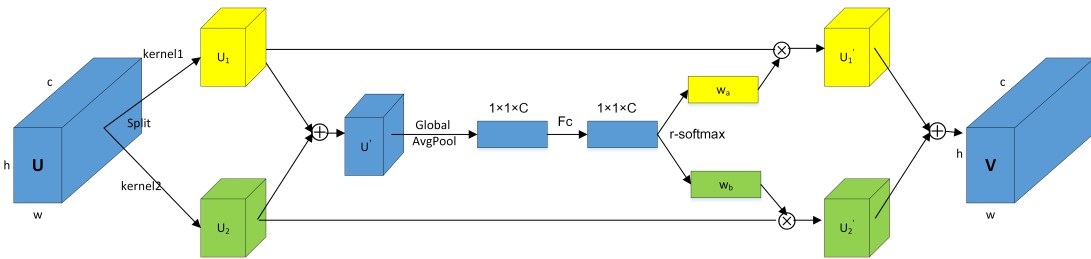

**Figure 3.** Illustration of the proposed in-channel attention block. Several channels are obtained by convolution, then calculate the weights of every channel and concatenate them together.

### 3.2. Pointcloud Feature Extraction

The 3D keypoints voting requires accurate pointcloud geometric reasoning and contextual information. The traditional approaches generally use a hand-crafted features such as the Fast Point Feature Histogram (FPFH), Signature of Histograms of Orientations Shot (SHOT), View point Feature Histogram (VFH) and others. These features are time-consuming and sensitive to noise, illumination, etc. PointNet++ [28] is the improved version of PointNet [13]. It proposes a hierarchical network to capture local features and uses PointNet to aggregate local neighborhood information. In addition, compared with the recent point cloud network PointCNN [44] and RandLA-Net [45], PointNet++ performs better on the Non-Uniform density point cloud due to the Multi-scale Grouping (MSG) and Multi-resolution grouping (MRG). So we choose it as the backbone. The proposed method handles the point cloud directly instead of transforming it into other structures.

This can avoid information loss during the processing. Meanwhile, due to the pointcloud's sparsity, the process is only for the interest points.

### 3.3. ORB-FPS Keypoint Selection

3D keypoint voting mainly includes two steps: select keypoints from the 3D model and design the keypoints prediction module to predict keypoints. Many of the former 3D key points methods are simple to select the corner points of the target's bounding box. However, the corner points are virtual and not on the object, which is not good for 6D pose estimation. PVN3D proposes Farthest Point Sample (FPS) to select the keypoints, but it only relies on the Euclidean distance, which may cause the keypoints to appear in areas without conspicuous features, such as a smooth surface. In order to avoid this, this paper proposes ORB-FPS 3D keypoints, we first detect the 2D ORB keypoints in the projected image of the 3D model, then calculate the corresponding 3D points on the model through the camera internal parameters, and finally find the $M$ ORB-FPS keypoints through FPS algorithm. The ORB algorithm first obtains the multi-scale images, then detects the significant keypoints through the Feature from Accelerated Segment Test (FAST) method, it is fast and scale invariant. The ORB keypoint selection process can be described as follows:

(1) Choose a pixel $p$ from the image and suppose its gray value is $G_p$.

(2) Calculate the gray value difference between $p$ and its neighboring points. Set the threshold $T$ and consider the two points are different when the difference is greater than $T$.

(3) Compare $p$ with 16 neighboring points and consider $p$ is the keypoint if there are $n(n = 12)$ consecutive points that are different from this point.

After obtaining the ORB keypoints and its corresponding 3D points on the model, use FPS to find the 3D keypoints. Specifically, we first select the model's center point as the first point, then add a point farthest from all the selected key points in turn until $M(M = 8)$ keypoints are obtained. The detail of the algorithm can be described in Algorithm 1.

---

**Algorithm 1:** ORB-FPS keypoint algorithm.

    **Input:** 2D projection image **I** 3D model **P** with N points
    **Output:** M ORB-FPS keypoints $k_i|_{i=1}^{M}$

1   Camera internal parameters $C$;
2   Number of ORB keypoints **A**;
3   Initial Center keypoint $O$;
4   n : Number of ORB neighbor points n=16;
5   t: ORB threshold;
6   **for** *i = 1:A* **do**
7      Select pixel p and set its value as l
8      Compute the difference d between l and its neighbor points
9      **if** *d>t* **then**
10        i = i+1;
11        **if** *i>12* **then**
12          p is the ORB keypoint;
13        **end**
14      **end**
15   **end**
16   **for** *j = 1:M* **do**
17      Obtain the 3D correspondence ORB point set A through C;
18      **while** *The selected ORB-FPS keypoints is less than M* **do**
19        Select the farthest point to the selected point set in A;
20      **end**
21   **end**

---

### 3.4. 3D Keypoint Voting

Through DenseFusion, the information from the image and the point cloud are closely combined to obtain the semantic features of each pixel. Based on the RGB-D information and offset strategy, we proposes the 3D keypoints voting module. In detail, we calculate the offset from each point to the predicted keypoints and the ground truth keypoints, and use their difference as the regression parameter. Suppose that the total number of points is $N$ and the keypoints is $M$, the loss can be expressed as:

$$L_{keypoint} = \frac{1}{N} \sum_{i=1}^{N} \sum_{j=1}^{M} ||\Delta v_{i,j}|_x|| + ||\Delta v_{i,j}|_y|| + ||\Delta v_{i,j}|_z|| + ||\Delta v_{i,j}|_f||$$

$$\Delta v_{i,j} = v_{i,j} - \hat{v}_{i,j}$$

(4)

where $v_{i,j}$ and $\hat{v}_{i,j}$ are the offset from the points to the predicted keypoints and groundtruth keypoints, respectively. $.|_x, .|_y, .|_z, .|_f$ are the components of $.$, and $.|_x, .|_y, .|_z$ denotes the 3D coordinate and $.|_f$ is the extracted feature.

### 3.5. Instance Semantic Segmentation

In order to improve the framework's ability to deal with multi object problem in 6d pose estimation, we add the instance semantic segmentation module into the framework. The common method utilizes semantic segmentation architecture to obtain the regions of interest (RoI) containing only a single object, and then performs keypoint detection on the RoI area. Using predictive segmentation to extract global and local features to classify objects is conducive to keypoint location, and the addition of size information can also help the framework to distinguish objects with similar appearance but different sizes. Based on this property, the proposed architecture performs these two tasks at the same time. Given the comprehensive features of each pixel to predict their semantic label, the Focal loss [46] can be described as:

$$L_{seg} = -\alpha(1 - p_i)^\gamma log(p_i)$$

$$p_i = c_i * h_i$$

(5)

where $\alpha$ is the balance parameter, it is to balance the importance of positive and negative samples. $\gamma$ is the focusing parameter to adjust the decreasing rate of simple samples. In this paper, we set $\alpha = 0.25$ and $\gamma = 2$. $c_i$ and $h_i$ are the predicted confidence and groundtruth label of the $i_{th}$ point.

### 3.6. Center Point Voting

Except for the keypoint, the center point voting module is also utilized in this part. The 3D center point will not be blocked by other objects, so it can treated as an assistant of the instance semantic segmentation. Following the PVN3D, we use the center point module to distinguish the different instance objects in the multi-objects scene. The 3D center point can also be considered a keypoint, so the center point loss $L_{centerpoint}$ can be described as the $L_{keypoint}$ where $M = 1$.

The 3D keypoint prediction network, instance semantic segmantation network and center point voting network are trained jointly, and the loss function could be defined as:

$$Loss = \lambda_1 L_{keypoiont} + \lambda_2 Lseg + \lambda_3 L_{centerpoiont}$$

(6)

### 3.7. Pose Calculate

Given two data sets of keypoints, one from the $M$ predicted keypoints $[k_i]_{i=1}^{M}$ in the camera's coordinate system, and another from the $M$ ORB-FPS keypoints $[k_i']_{i=1}^{M}$ in the object's coordinate system. The least-square method is used to calculate 6d pose translation,

which is to find the best rotation matrix $R$ and translation vector $T$ that make the two datasets keypoints closest to each other. The $R$ and $T$ can be calculated as follows:

$$R, T = \arg \min_{R,T} \sum_{i=1}^{M} ||k_i - (R * k'_i + t)||^2 \qquad (7)$$

## 4. Experiments

This section shows the experimental results of the proposed method. We evaluate the performance with other pose estimation algorithms on two datasets, Linemod dataset [47] and YCB-Video dataset [9].

### 4.1. Datasets

Linemod is a standard benchmark widely used in 6d pose estimation. Linemod consists of 13 objects from 13 videos. This dataset contains multiple challenging problems for pose estimation: cluttered scenes, texture-less objects and different lighting conditions.

YCB-Video Dataset is used to evaluate the pose estimation performance in symmetry and severe occlusions. The dataset contains 21 YCB objects with varying shapes and textures and captures and annotates the 6D pose from 92 RGBD videos. Follow PoseCNN, we split the dataset into 80 videos for training and another 12 videos for testing. Meantime, we add the synthetic images into the training set.

### 4.2. Evaluation Metrics

We evaluate our framework with the ADD and ADD-S metrics. The ADD evaluates the average Euclidean distance between the model points transformed with the predicted 6D pose $[R, t]$ and the groudtruth pose $[\hat{R}, \hat{t}]$:

$$\text{ADD} = \frac{1}{N} \sum_{x \in O} ||(Rx + t) - (\hat{R}x + \hat{t})|| \qquad (8)$$

where $O$ is the object mesh and the $N$ is the total number. In addition, For symmetrical objects like eggbox and glue, the ADD-S is proposed. ADD-S computes the distance with the closest points. The formula can be described as follows:

$$\text{ADD} - \text{S} = \frac{1}{N} \sum_{x_1 \in O} \min_{x_2 \in O} ||(Rx_1 + t) - (\hat{R}x_2 + \hat{t})|| \qquad (9)$$

We follow PoseCNN [9] and report the area under the ADD-S curve (AUC) obtained by changing the average distance threshold as the evaluation. Furthermore, in this paper we set the max accuracy threshold to 2cm.

### 4.3. Implementation Details

During implementation, follow the PVN3D, we select eight ORB-FPS keypoints and one center point for the predicting network. Furthermore, during the training and testing processes, we sample 12,888 points for each frame of RGBD image. In order to enhance the framework's robustness to the light condition and background, we synthesize 70,000 rendered images and 10,000 fused multi-objects images from the SUN2012pascalformat dataset [48]. In addition, we set the initial learning rate 0.001 and decrease 0.00001 every 20,000 iterations.

### 4.4. Results on the Benchmark Datasets

In order to evaluate the pose estimation performance of our algorithm, we design several group of experiments and show the experiment results. The ADD(S) metric means ADD metric for the non-symmetrical objects and ADD-S for the symmetrical objects.

### 4.4.1. Result on the Linemod Dataset

Table 1 shows our quantitative evaluation results of the 6D pose estimation experiments on the Linemod dataset. We compare our method with the RGB based methods PoseCNN [9], PVNet [7] and RGDB based methods PointFusion [49], Densefusion [12], PVN3D [15]. Among them, we reference PoseCNN, PVNet, PointFusion and DenseFusion as the baseline. For PVN3D, they release their pretrained model on the Linemod and YCB datasets, so we test the performance of the pretrained model on the datasets. These methods are the single view and without iterative refinement. As the table shows, the BB8 means 8 bounding box points and it makes the worst performance due to the keypoints may not appear on the surface of the object. In addition, the RGBD based methods perform better than RGB based methods demonstrates that RGB and point cloud features are necessary for this task. Furthermore, the proposed algorithm advances the other methods by more than 0.3 percent on the ADD(S) metric due to its better representation ability and the keypoint selection scheme. Figure 4 shows the visualization results of our approach on the Linemod dataset, it can be seen that our method performs well on the Linemod objects.

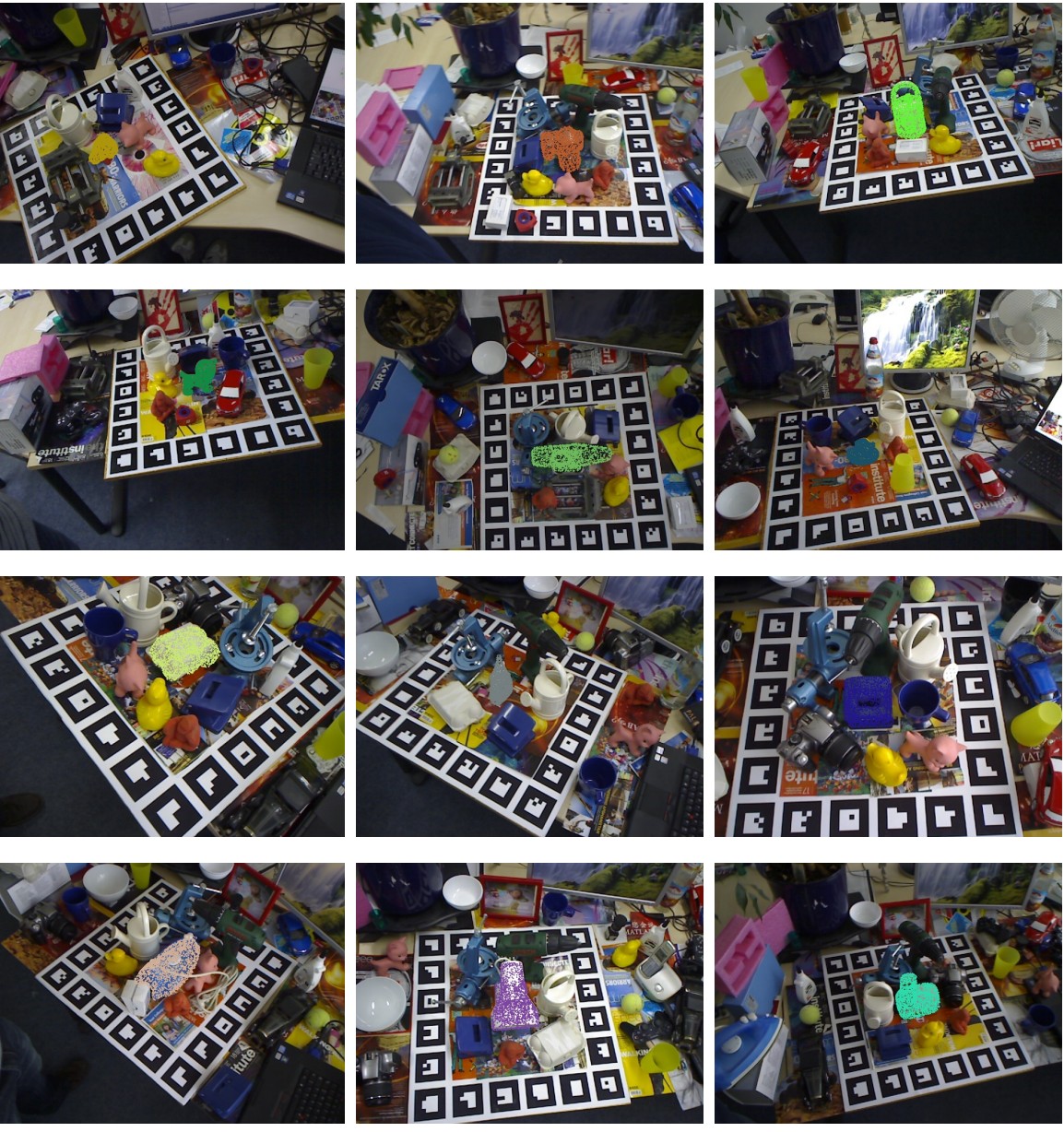

**Figure 4.** Some visual results of the Linemod dataset.

**Table 1.** The accuracies of our method and the baseline methods on the LINEMOD dataset in terms of the ADD(-S) metric, where the bold are considered as symmetrical objects.

|  | PoseCNN | PVNet | PointFusion | Densefusion | PVN3D | Ours |
|---|---|---|---|---|---|---|
| ape | 77.0 | 43.6 | 70.4 | 79.5 | 96.0 | 96.4 |
| benchvise | 97.5 | 99.9 | 80.7 | 84.2 | 94.2 | 94.5 |
| cam | 93.5 | 86.9 | 60.8 | 76.5 | 92.3 | 95.0 |
| can | 96.5 | 95.5 | 61.1 | 86.6 | 93.8 | 94.9 |
| cat | 82.1 | 79.3 | 79.1 | 88.8 | 95.4 | 95.8 |
| driller | 95 | 96.4 | 47.3 | 77.7 | 93.0 | 94.0 |
| duck | 77.7 | 52.6 | 63.0 | 76.3 | 93.7 | 92.9 |
| **eggbox** | 97.1 | 99.2 | 99.9 | 99.9 | 96.2 | 96.7 |
| **glue** | 99.4 | 95.7 | 99.3 | 99.4 | 96.4 | 97.4 |
| holepuncher | 52.8 | 81.9 | 71.8 | 79.0 | 94.5 | 96.3 |
| iron | 98.3 | 98.9 | 83.2 | 92.1 | 92.2 | 93.1 |
| lamp | 97.5 | 99.3 | 62.3 | 92.3 | 93.5 | 93.7 |
| phone | 87.7 | 92.4 | 78.8 | 88.0 | 93.5 | 93.6 |
| average | 88.6 | 86.3 | 73.7 | 86.2 | 94.2 | **94.5** |

4.4.2. Result on the YCB-Dataset

Table 2 shows the quantitative evaluation results of the 6D pose on the YCB-Video dataset of the proposed algorithm and other methods. The compared methods are PoseCNN, DenseFusion and PVN3D and all the methods do not have iterative refinement. As the Table 2 shows, our method advances other approaches by 1.7 percent on the ADD-S metric and 0.8 percent on the ADD(S) metric. Our method also outperforms than others on the symmetrical objects (tagged in bold in the table). Figure 5 shows the visualization results of our method on the YCB-Video dataset, some of them are severely occluded and the performances of the proposed method are excellent as well. This is because our algorithm uses dense prediction, which calculates every point/pixel's translation vector to the keypoint and votes the predicted keypoints through the vectors. The voting scheme is motivated by the property of the rigid objects, which means once we see the local visible parts, we can infer the relative directions to the invisible parts. Furthermore, the comprehensive utilization of RGB images and point cloud leads to better performance on the symmetrical object. It can be concluded that our algorithm works well on the occlusion scenes.

**Table 2.** Quantitative evaluation of 6D Pose (ADD-S, ADD(S)) on the YCB dataset. Symmetric objects' names are in bold.

|  | PoseCNN | | DenFusion | | PVN3D | | Ours | |
|---|---|---|---|---|---|---|---|---|
|  | ADD-S | ADD(s) | ADD-S | ADD(s) | ADD-S | ADD(s) | ADD-S | ADD(s) |
| 002 master chef can | 83.9 | 50.2 | 95.3 | 70.7 | 95.8 | 79.6 | 96.4 | 81 |
| 003 cracker box | 76.9 | 53.1 | 92.5 | 86.9 | 95.4 | 93.0 | 94.5 | 90.0 |
| 004 sugar box | 84.2 | 68.4 | 95.1 | 90.8 | 97.2 | 95.9 | 97.1 | 95.6 |
| 005 tomato soup can | 81 | 66.2 | 93.8 | 84.7 | 95.7 | 89.8 | 95.6 | 89.2 |
| 006 mustard bottle | 90.4 | 81 | 95.8 | 90.9 | 97.6 | 96.5 | 97.6 | 95.4 |
| 007 tuna fish can | 88 | 70.7 | 95.7 | 79.6 | 96.7 | 91.7 | 96.7 | 87.0 |
| 008 pudding box | 79.1 | 62.7 | 94.3 | 89.3 | 96.5 | 93.6 | 95.1 | 90.6 |
| 009 gelatin box | 87.2 | 75.2 | 97.2 | 95.8 | 97.4 | 95.1 | 97.2 | 95.6 |
| 010 potted meat can | 78.5 | 59.5 | 89.3 | 79.6 | 92.1 | 84.4 | 91.2 | 87.1 |
| 011 banana | 86 | 72.3 | 90 | 76.7 | 96.3 | 92.4 | 96.9 | 94.1 |
| 019 pitcher base | 77 | 53.3 | 93.6 | 87.1 | 96.2 | 93.8 | 96.9 | 95.5 |
| 021 bleach cleanser | 71.6 | 50.3 | 94.4 | 87.5 | 95.5 | 90.9 | 96.1 | 93.4 |

**Table 2.** *Cont.*

|  | PoseCNN | | DenFusion | | PVN3D | | Ours | |
|---|---|---|---|---|---|---|---|---|
|  | **ADD-S** | **ADD(s)** | **ADD-S** | **ADD(s)** | **ADD-S** | **ADD(s)** | **ADD-S** | **ADD(s)** |
| **024 bowl** | 69.6 | 69.6 | 86 | 86 | 85.5 | 85.5 | 95.4 | 95.4 |
| 025 mug | 78.2 | 58.5 | 95.3 | 83.8 | 97.1 | 94.0 | 97.3 | 92.8 |
| 035 power drill | 72.7 | 55.3 | 92.1 | 83.7 | 96.6 | 95.0 | 96.7 | 94.9 |
| **036 wood block** | 64.3 | 64.3 | 89.5 | 89.5 | 90.8 | 90.8 | 95.1 | 95.1 |
| 037 scissors | 56.9 | 35.8 | 90.1 | 77.4 | 91.8 | 91.8 | 96.5 | 92.6 |
| 040 large marker | 71.7 | 58.3 | 95.1 | 89.1 | 95.2 | 90.1 | 94.3 | 86.2 |
| **051 large clamp** | 50.2 | 50.2 | 71.5 | 71.5 | 90.0 | 90.0 | 95.4 | 95.4 |
| **052 extra large clamp** | 44.1 | 44.1 | 70.2 | 70.2 | 77.6 | 77.6 | 95.5 | 95.5 |
| **061 foam brick** | 88 | 88 | 92.2 | 92.2 | 95.4 | 95.4 | 96.5 | 96.5 |
| Average | 75.2 | 61.3 | 90.9 | 82.9 | 94.2 | 90.7 | **95.8** | **91.4** |

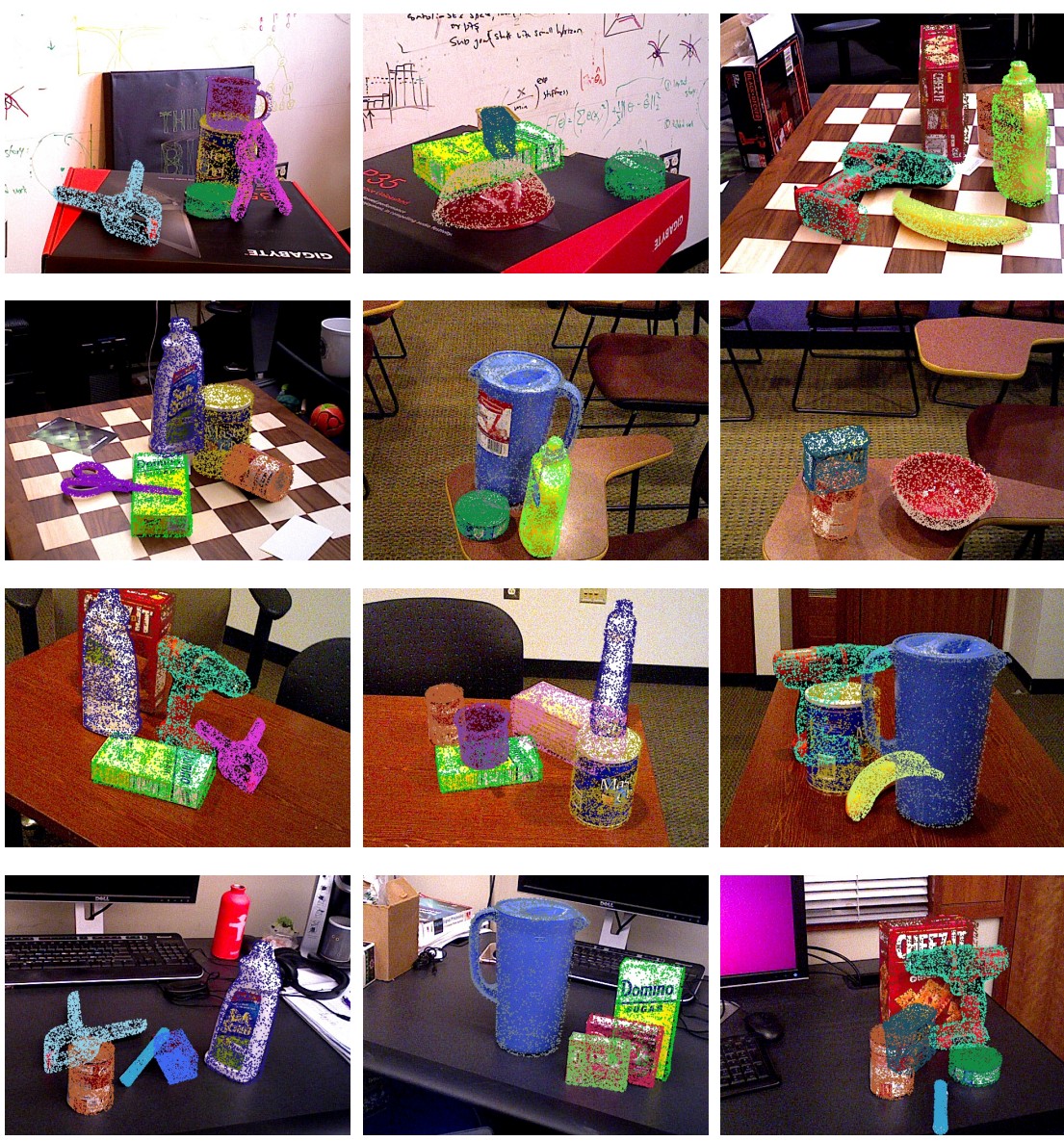

**Figure 5.** Some visualization results of the YCB dataset. The proposed method performs well even when the objects are seriously occluded.

*4.5. Ablation Study*

A series of ablation studies are conducted to analyse the influences of different keypoints selection methods, number of keypoints on the framework and the in-channel weight calculation.

**Effect on the Keypoint Selection.** In this part, we select 8 corner points of the 3D object's bounding box(BB8) and 8 Farthest Point Sample(FPS) points to compare with our ORB-FPS keypoint. In addition, the different number of ORB-FPS keypoints are also taken into the comparison. Table 3 shows the experiment result of the several methods on the YCB-dataset. Overall, the ORB-FPS keypoint selection scheme performs better than the FPS on the basics of selecting the same number of key points, this is because ORB-FPS makes better use of the RGB image's texture feature. The BB8 keypoint performs worst due to the bounding box corner points are virtual and far away from the object. In addition, from the comparison of the different number of keypoints, eight keypoints selected from the ORB-FPS approach is a good choice. Fewer keypoints can not fully express the complete shape of the object and more keypoints will increase the output space and make the network harder to learn.

**Table 3.** Experiment results on YCB dataset of the different keypoint selection approaches.

|        | BB8  | FPS8 | ORB-FPS4 | ORB-FPS8 | ORB-FPS12 |
|--------|------|------|----------|----------|-----------|
| ADD-S  | 93.2 | 94.2 | 94.1     | 95.8     | 94.7      |
| ADD(S) | 89.4 | 90.7 | 90.5     | 91.4     | 91.0      |

**Effect of the channel-attention.** In order to testify the influence of the channel-attention, we compare the experimental results with the channel-attention and without. According to Table 4, adding channel-attention block increase the 6D pose performance. The channel-attention block can calculate the weight of different channels and use these weights to integrate the channel-wise information, this will increase the network's representation ability.

**Table 4.** Experiment results on YCB dataset with/withou the channel-attention.

|        | Without | With |
|--------|---------|------|
| ADD-S  | 95.6    | 95.8 |
| ADD(S) | 91.0    | 91.4 |

## 5. Conclusions

This paper presents an accurate deep 3D keypoint voting network for 6DoF pose estimation. We propose split-attention block to enhance the network's ability to learn features from the RGB image. Due to the split channel attention, the network can selectively enhance useful channels and suppress less useful ones. In addition, we introduce the 3D ORB-FPS keypoint selection method, which first detects the 2D ORB keypoints in the image, then finds the corresponding 3D points through the camera intrinsic matrix, and finally finds the 3D keypoints through the FPS algorithm. The proposed keypoint selection method leverages texture information of RGB image and geometry information of pointcloud. Our algorithm outperforms the previous methods on two benchmark datasets on the ADD-S and ADD(S) metrics. Overall, we believe our network can be applied in the real applications such as Automatic driving, Bin-Picking and so on.

**Author Contributions:** Conceptualization, S.S.; methodology, S.S. and H.L.; software, H.L., Y.Z. and L.L.; validation, S.S., G.L. and H.L.; formal analysis, H.L. and H.X.; investigation, S.S.; resources, S.S.; data curation, H.L. and Y.L.; writing original draft preparation, H.L.; writing review and editing, H.L. and S.S.; visualization, H.L. and H.X.; supervision, S.S.; project administration, S.S. and H.L.; funding acquisition, S.S. All authors have read and agreed to the published version of the manuscript.

**Funding:** This research was funded by the Innovation Project of Shanghai Institute of Technical Physics, Chinese Academy of Sciences (NO. X-209).

**Data Availability Statement:** Not applicable.

**Acknowledgments:** The authors would like to acknowledge the Linemod dataset and the YCB dataset for making their datasets available to us.

**Conflicts of Interest:** The authors declare no conflict of interest.

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
