# Peer review of "A 3D Keypoints Voting Network for 6DoF Pose Estimation in Indoor Scene"

_machines, doi:10.3390/machines9100230_

Round 1

Reviewer 1 Report

The presented work introduces a deep 3D keypoint voting network for 6DoF pose estimation.  The main contribution is a split-attention block to refine features from the RGB image. In addition the authors introduce a 3D ORB-FPS keypoint selection method leveraging texture information from RGB image and geometric information form the point cloud. The paper is well-written and well motivated, as well as the results and comparison are convincing.

As presented the work seems already well reviewed in its parts, however the related works section needs to be completed in the paragraph related to pose from RGB images (2.1 - see references below). Specifically, authors stated that current image-based methods   are targeted to perspective projection,  which may cause the partly loss of the geometric constraint information.

Actually, since the problem of the limited field of view is common to many reconstruction and scene understanding tasks, an important branch of research in this field has focused on the use of panoramic images indoors (i.e., which have become a common standard also for rgbd acquisitions). Starting from this context it is worth mentioning the work of Zhang et al.[Zhang:PAN:2014], where they show how the use of panoramic images improves significantly the geometric analysis of the scene thanks to the large captured context, the work of Xu et al.[Xu:2017:P2C], which extends the work of Zhang et al.[Zhang:PAN:2014] with object poses, and the work of Pintore et al.[Pintore:2019:AMC], which combines modern CNN networks, 3D scene information and parameter optimization to recover the oriented bounding boxes of the captured objects inside the 3D layout of the indoor environment. Obviously since these papers are orthogonal to the submitted work, no further comparison is necessary.

@InProceedings{Zhang:PAN:2014,

author="Zhang, Yinda and Song, Shuran and Tan, Ping and Xiao, Jianxiong",

title="PanoContext: A Whole-Room 3D Context Model for Panoramic Scene Understanding",

booktitle="Proc. ECCV",

year="2014",

pages="668--686",

}

@INPROCEEDINGS{Xu:2017:P2C,

author={J. {Xu} and B. {Stenger} and T. {Kerola} and T. {Tung}},

booktitle={Proc. WACV},

title={{Pano2CAD}: Room Layout from a Single Panorama Image},

year={2017},

pages={354-362},

}

@Article{Pintore:2019:AMC,

    author = {Giovanni Pintore and Fabio Ganovelli and Alberto {Jaspe Villanueva} and Enrico Gobbetti},

    title = {Automatic modeling of cluttered multi-room floor plans from panoramic images},

    journal = {Computers Graphics Forum},

    volume = {38},

    number = {7},

    pages = {347--358},

    year = {2019},

}

Reviewer 2 Report

This manuscript extracts feature points from both RGB and 3D data, and use them for pose estimation. Several existing deep learning networks are included for network design and research with RGB-D used as input. As at least one version has been revised when reviewer gets this manuscript, the comments are made based on the black and blue contents.

1.1 line39-40: It is said ‘ICP is extremely time-consuming and cannot be optimized end-to-end.’ The complexity of ICP should not be larger than most deep learning methods. Considering the point number is not quite large, ICP should be very fast on testing data.

2.1 Can traditional feature extraction method find correspondence for RGB images? Such as SIFT. As the images used here have quite distinct and obvious features.

3.2 line: 194-204, The principle of PointNet++ is not introduced, only the comparison between traditional methods and PointNet++ is explained. The author does not involve the optimization of the latest network selection, as pointnet++ (proposed 4 years ago) is not the only model handling point directly, e.g., Randnet, PointCNN. Reviewer would recommend the authors add the reason that PointNet++ is used here.

4.4 Are the results of the compared methods (PoseCNN, PoseCNN and PVN3D) generated by the authors or referenced as baseline? If those are run by authors, suggest adding the parameters.

4.4 Results in 4.4.1 and 4.4.2 seem quite simple. Suggest authors add some discussions to further explain the advantages detailedly, such as the results on symmetric and occluded objects in terms of Fig.5.

Please check the writing.

Line 62: ‘are calculate’->’are calculated’

Line101-104: Please revise those sentences.

Line 287: ‘severe occluded’-> ‘severely occluded’?

Line 316 ‘method’->’methods’?

Reviewer 3 Report

  • Some reference numbers are preceded by a space, others are not.
    Some figures have unnecessarily long titles and are not referenced in the text.
    Subchapter headings follow immediately after chapter headings in some cases. In my opinion, it would be useful to add a short descriptive paragraph as in the case of Chapter 4.4.
    several parts of the text problematic from a linguistic point of view have been corrected, but there are still some minor shortcomings (e.g. line 285).

Round 2

Reviewer 2 Report

My concerns related with the scientific content itself have been addressed. I still recommend the authors revise the writing again, e.g., please check the revised sentence line277-279 as it seems not a complete description.

Author Response

This manuscript is a resubmission of an earlier submission. The following is a list of the peer review reports and author responses from that submission.

Round 1

Reviewer 1 Report

6 DoF pose estimation is a fundamental task in computer vision. The authors review the drawbacks of related RGB image-based and depth-/point cloud-based methods to conclude that the combination of RGB and depth information should improve the performance of 6 DoF pose estimation. Based on this motivation, authors propose a novel network to validate their ideas. The paper is well written, and I would like to suggest a major revision for this submission. I mainly have the following concerns:

  • Page 3, regarding “[…] From the above discussion, a conclusion can be made that effective comprehensive utilization of image and point cloud information can improve the network’s ability to perform pose estimation […]”. Despite I this statement should be correct; the given evidence is not strong to support this conclusion. Authors should pay an attention to the logic to naturally build a bridge to this conclusion from these references.
  • Please give the full name or definition for the ORB key point.
  • The proposed method takes RGB images and depth images/point clouds as input. The works of feature fusion for RGB images and depth images/point clouds are missing. Authors need a subsection under Related Work about RGB+depth/point clouds feature fusion. For example, authors may need to cite the papers “Performance evaluation of deep feature learning for RGB-D image/video classification”, “Multi‐modal deep network for RGB‐D segmentation of clothes”, “RGB and lidar fusion based 3d semantic segmentation for autonomous driving”.
  • In the proposed method, authors use PointNet++ to extract features of point clouds. PointNet++ indeed performs better than PointNet in terms of segmentation and classification. However, PointNet has been proved to perform better than PointNet++ in other tasks (e.g. 3D shape reconstruction). Therefore, I suggest the authors should compare the performance of PointNet in their method.
  • In this article, the authors use DenseFusion proposed by ref [11] to fuse the features of RGB images and point clouds. I am also interested in seeing the comparisons of difference feature fusion strategies. Does fusion of RGB images and depth images perform better than the fusion of RGB images and point clouds?

Reviewer 2 Report

  1. The manuscript designs the 3D keypoint voting network (3DKV) for 6DOF pose estimation, 3DKV is a two-stage network, first detects the keypoints, then utilizes the least squares method to compute the pose. In fact, the 3DKV algorithm described in the manuscript is also relatively conventional and does not have much innovation. On the contrary, it may increase the amount of calculation and increase much more processing steps, but the actual effect improvement is very small and not obvious compared with PVN3D method.

In addition, the existing 6DOF pose estimation methods, such as some methods in industrial photogrammetry, have very high calculation accuracy, which is higher than that in this manuscript.

  1. The ADD evaluation metrics in this manuscript is used to evaluate the accuracy of the predicted 6D pose. For some special cases, the evaluation index has large problems and obvious loopholes, which is unscientific.

It is suggested that the author use some measured control points (GCP) to test the accuracy of he predicted 6D pose. For example, control points are used to calculate the accuracy of aerial triangulation (Aerotriangulation). Such accuracy evaluation method is much more scientific, more accurate and more practical.

  1. It is suggested to limit the title of the manuscript. What scene the algorithm is aimed at and what data is it based on? Otherwise, it is easy to mislead readers.

  1. There are some writing.problems of the abstract. For example:

(1) It is recommended to write abstracts from a third person perspective rather than "we".

(2) There are too many background introductions, but the author does not clearly point out the specific targeted problems of the manuscript, and the descriptions are too broad.

(3) In addition, the abstract is basically an abbreviated small paper, including multiple elements. However, the description of the experiment and experimental conclusions in the manuscript is relatively simple and not specific enough, and the experimental effect needs to be quantified as much as possible.

  1. In the abstract, "most of the methods have not made the best use of the texture information of the RGB image and the geometry information of the depth image" is inappropriate. At present, many good pose estimation and 3D reconstruction algorithms that fully consider the texture information of the RGB image and the geometry information of the depth image. Moreover, RGBD sensor is not suitable for long-distance observation or some outdoor scenes. Therefore, for different scenarios and requirements, there should be different sensor combinations, which should not be generalized.

(1) What scene needs to use RGB image and depth map image at the same time? And please clarify: What scene the algorithm is aimed at (the specific targeted scenario of the manuscript)?

(2) In addition, the targeted problem should be one or two specific points. Please make the targeted problem in this article more specific and not too general. If it is too general, it means that you have not understood this research thoroughly and have not mastered this field deeply, or deliberately exaggerate it.

  1. The 3D ORB-FPS keypoint selection method described in this paper is not very new and innovative. Based on the orb matching algorithm, it combines the Farthest Point Sample (FPS) method to find the final keypoints. However, the Farthest Point Sample (FPS) method is not proposed by the author, so it is not innovative compared with the current method, and it doesn't have much practical improvement.

  1. What effect does the FPS method described in the manuscript have on the ORB feature matching? In addition to the improvements, what bad effects and deficiencies have also been brought (for example, the adaptability to some special types of data has become worse, and the stability and robustness of the algorithm have become worse)?

  1. In the experiment, the number of videos used for training is too large, but the number of videos used to test the algorithm is too small.

  1. How does this article convert ADD evaluation metric result (distance) into percentage accuracy? The author did not elaborate in his manuscript. And whether the author's conversion method is scientific or not?

  1. The manuscript combines the existing methods, and it increases the computational complexity of the method and adds much more processing steps. However, from the experimental results, the actual effect improvement of this manuscript is very small and not obvious compared with the existing PVN3D methods.

  1. Please compare the computational complexity of this method with the existing methods in the manuscript.

In addition, please increase the computational efficiency comparison experiment between this method and the existing methods, such as PoseCNN, PVNet, PointFusion, Densefusion, PVN3D, et al.

  1. Some references are not in the correct format. For example, the references 8, 10, 13, 16-18, et al.

Round 2

Reviewer 1 Report

The authors have addressed most of my concerns. I, thus, suggest accepting this submission.

Reviewer 2 Report

  1. Point 1: The manuscript designs the 3D keypoint voting network (3DKV) for 6DOF pose estimation, 3DKV is a two-stage network, first detects the keypoints, then utilizes the least squares method to compute the pose. In fact, the 3DKV algorithm described in the manuscript is also relatively conventional and does not have much innovation. On the contrary, it may increase the amount of calculation and increase much more processing steps, but the actual effect improvement is very small and not obvious compared with PVN3D method.

In addition, the existing 6DOF pose estimation methods, such as some methods in industrial photogrammetry, have very high calculation accuracy, which is higher than that in this manuscript.

The author replied that “Thank you very much for some methods in industrial photogrammetry mentioned by the reviewer. We found that these methods have very high calculation accuracy, but they rely on accurate measuring instruments. Our acquisition equipment is Kinect, it is low-cost and not very accurate, so these methods are not very suitable. We really hope that the corrections will meet with your approval. Special thanks to you for your good comments again.” However, some method is just need RGB images, and is much cheap than the Kinect. Thus, the authors’ presentation is not correct.

  1. Point 2: The ADD evaluation metrics in this manuscript is used to evaluate the accuracy of the predicted 6D pose. For some special cases, the evaluation index has large problems and obvious loopholes, which is unscientific.

It is suggested that the author use some measured control points (GCP) to test the accuracy of he predicted 6D pose. For example, control points are used to calculate the accuracy of aerial triangulation (Aerotriangulation). Such accuracy evaluation method is much more scientific, more accurate and more practical.

The author replied that “the control point method will not be suitable for the indoor scene” is not correct!

  1. Point 5: In the abstract, "most of the methods have not made the best use of the texture information of the RGB image and the geometry information of the depth image" is inappropriate. At present, many good pose estimation and 3D reconstruction algorithms that fully consider the texture information of the RGB image and the geometry information of the depth image. Moreover, RGBD sensor is not suitable for long-distance observation or some outdoor scenes. Therefore, for different scenarios and requirements, there should be different sensor combinations, which should not be generalized.

(1) What scene needs to use RGB image and depth map image at the same time? And please clarify: What scene the algorithm is aimed at (the specific targeted scenario of the manuscript)?

(2) In addition, the targeted problem should be one or two specific points. Please make the targeted problem in this article more specific and not too general. If it is too general, it means that you have not understood this research thoroughly and have not mastered this field deeply, or deliberately exaggerate it.

The author replied that “The indoor scene is complex, and there may be illumination change and occlusion problems.” is no meaningful, because many scenes are complex, and there are illumination change and occlusion problems.

And the authors reply of “RGB image has rich texture information but not geometry information” is wrong, because the stereo image pairs (two or serval RGB images) can restore geometry information!

  1. Point 6: The 3D ORB-FPS keypoint selection method described in this paper is not very new and innovative. Based on the orb matching algorithm, it combines the Farthest Point Sample (FPS) method to find the final keypoints. However, the Farthest Point Sample (FPS) method is not proposed by the author, so it is not innovative compared with the current method, and it doesn't have much practical improvement.

The author replied that “We appreciate the reviewer's thoughtful comments and thank you very much for the precious comments. Actually, we found that the previous algorithm chooses the keypoints through the Farthest Point Sample (FPS) method. However, this has a drawback that some selected keypoints may appear in non-salient regions like smooth surfaces without distinctive texture, making them hard to locate. Instead, firstly we detect the ORB keypoint in the RGB images, then calculate the correspondence 3D points in the point cloud through the camera parameter, then find the final ORB-FPS keypoints through FPS from the selected point. This will solve this problem. We really hope that the corrections will meet with your approval. Special thanks to you for your good comments again.”

However, the authors’ method is not very new and innovative, and the method is very easy and the modification is small, simple and very common practice.

  1. The manuscript combines the existing methods, and it increases the computational complexity of the method and adds much more processing steps. However, from the experimental results, the actual effect improvement of this manuscript is very small and not obvious compared with the existing PVN3D methods.

It’s not a very meaningful method.

  1. Point 11: Please compare the computational complexity of this method with the existing methods in the manuscript.

To compare the computational complexity of these method, is not just to compare the number of parameters.